# Proneurogenic Actions of FSH During Directed Differentiation of Neural Stem and Progenitor Cells from Ovarian Cortical Cells Towards the Dopaminergic Pathway

**DOI:** 10.3390/biomedicines13071560

**Published:** 2025-06-26

**Authors:** Alfredo González-Gil, Concepción Rojo, Esther Ramírez, Ricardo Martín, Alberto Samuel Suárez-Pinilla, Susana Ovalle, Ricardo Ramos-Ruiz, Rosa Ana Picazo

**Affiliations:** 1Department of Physiology, School of Veterinary Medicine, Complutense University of Madrid, 28040 Madrid, Spain; alfgonza@ucm.es (A.G.-G.); esthrami@ucm.es (E.R.); 2Department of Anatomy and Embryology, School of Veterinary Medicine, University Complutense of Madrid, 28040 Madrid, Spain; rojosalv@vet.ucm.es; 3Department of Physiology, School of Medicine, Complutense University of Madrid, 28040 Madrid, Spain; martinh@ucm.es; 4Institute of Health Research, Hospital Clínico San Carlos, 28040 Madrid, Spain; albersua@ucm.es; 5Department of Biochemistry and Molecular Biology, School of Veterinary Medicine, Complutense University of Madrid, 28040 Madrid, Spain; 6University Institute for Research in Neurochemistry, Complutense University of Madrid, 28040 Madrid, Spain; 7Genomic Unit Cantoblanco, Fundación Parque Científico de Madrid, Madrid 28049, Spain; susana.ovalle@fpcm.es; 8IMDEA Research Institute on Food and Health Science, Ctra. Madrid 28049, Spain; ricardo.ramos@alimentacion.imdea.org

**Keywords:** FSH, neurogenesis, dopaminergic neuron, neural stem and progenitor cells, ovarian cortical cells

## Abstract

Exploring the neurogenic potential of extraneural stem cells under the actions of proneurogenic biomolecules may enhance the success of autologous cell therapy for neurodegenerative diseases such as Parkinson’s. Neural stem and progenitor cells (NSPCs) from extraneural tissues have emerged as potential sources of functional dopaminergic (DA) neurons. **Background/Objectives**: This study aimed to generate DA neurons from ovarian cortical cells (OCC)-derived NSPCs to elucidate whether follicle-stimulating hormone (FSH) can enhance this process and to evaluate the electrophysiological functionality of differentiated neural cells using the patch-clamp technique. **Methods**: OCC-NSPCs were differentiated towards the DA pathway during the neurosphere (NS) assay after two culture periods for cell expansion (CEP-1, CEP-2) with one of these media: M1 (positive control with epidermal growth factor, EGF, and fibroblast growth factor2, FGF2), M2 (control), and M3 (M2 with FSH, 50 ng/mL). Image analysis, morphometric evaluation, cell proliferation assays, and gene expression analysis of NSPC-specific transcripts were performed. After CEP-2, NS cells were cultured for 30 days in a serum-free medium containing Sonic-Hedgehog, FGF2, FGF8, and brain-derived neurotrophic factor (BDNF) for differentiation. At the end of culture, expression, and immunolocalization of GFAP, Olig2, NeuN, and tyrosine hydroxylase (TH) were analyzed in cells, along with patch-clamp recordings in differentiated neurons. **Results**: Cell proliferation and NS development were larger in OCC-NSPCs from groups M1 and M3 than in M2. Expression of NSPC-related transcripts was higher in M2; however, M1 and M3 cultures showed greater expression of differentiation markers *NeuN*, *GFAP*, *Olig2*, and *TH*. NeuN, GFAP, and TH were immunolocalized in differentiated cells and NS that were generated during differentiation. TH was localized in neural precursor cells, some neurons, core cells of small-, medium-, and large-sized NS, and in cells close to the outer cell layer of large NS, with greatest immunolocalization percentages in NS primed with FSH during CEP-1/2 (M3). Electrophysiological recordings revealed a major incidence of plateau potentials and a significant proportion of complete action potentials, reflecting successful functional neuronal differentiation. **Conclusions**: DA precursors and functional neurons can be successfully obtained after OCC-NSPCs-directed differentiation. FSH priming during the expansion period enhances the neurogenic potential of these cells towards the DA pathway. Future research will explore the eventual therapeutic use of these findings for neurodegenerative diseases.

## 1. Introduction

Advances in stem cell (SC) research have unveiled their therapeutic potential for replacing damaged cells and repairing injured tissues, both in regenerative medicine and in neurodegenerative diseases [1]. Parkinson’s disease (PD) is a progressive neurodegenerative disorder characterized by resting tremor, rigidity, bradykinesia, and gait disability, resulting from the loss of dopaminergic (DA) neurons in the *substantia nigra pars compacta* [2,3,4]. Most current treatments for PD focus on DA cell replacement therapies in the affected areas. These approaches, which involve transplanting DA neurons derived from pluripotent SCs, are currently under clinical trials [5].

Despite the promising potential of pluripotent SCs in regenerative medicine, there is a need to generate patient-specific SCs without ethical, immunological, and genetic instability concerns, providing an easily available and renewable source of cells with the potential to differentiate into fully functional DA neurons after transplantation [6]. In this context, adult SCs from non-central nervous system (CNS) tissues are emerging as a reliable alternative, offering advantages such as the absence of ethical or legal constraints, accessibility from body tissues, and the possibility of autologous use, thus preventing immune rejection [7,8]. Furthermore, these cells do not exhibit the tumorigenic risks associated with the use of induced pluripotent stem cells (iPSC) as observed in animal models [9]. Nevertheless, assessing tumorigenicity remains a critical step in ensuring the safety of SC-based therapies and is currently the focus of ongoing basic research preceding preclinical trials [9,10]. Neural stem cells (NSCs) have been isolated from adult SCs harbored in extraneural tissues, such as the hair follicle [11], dental pulp [12], muscle [13], and ovarian cortical tissue [14,15,16,17]. Until recently, there was no strong evidence that these SCs could differentiate into functional DA neurons [2]. However, recent studies have revealed that in vitro differentiation towards DA neurons has been possible from adult SCs of adipose tissue [18], dental pulp [19], and testes [20].

The ability of extraneural SCs to differentiate into functional DA neurons may have important consequences for PD therapy, as it could provide an ethically uncontroversial and inexhaustible source of donor cells for use in models of neurodegenerative diseases and in autologous cell therapy trials [2].

The ovarian surface epithelium (OSE) in mammals contains two types of SC populations: very small embryonic-like SCs and ovarian germ SCs [21]. Both OSE and stromal pluripotent cells have demonstrated the ability to differentiate into cells expressing characteristic markers of neural cells in culture, a process promoted by sex steroids [22]. Moreover, a recent study described a culture system supporting the spontaneous generation of aggregates or three-dimensional structures, called neurospheres (NS), from sheep ovarian cortical tissue cells. These NS-derived cells exhibit self-renewal capabilities and the potential to differentiate into both neurons and glial cells [14]. The use of NS allows for a more accurate reproduction of the natural and functional microenvironment in which these cells are found in vivo [23], thereby enabling the generation of new neurons with potential applications for neurodegenerative diseases.

The analysis of the differentiation potential of neural stem and progenitor cells (NSPCs) towards specific neuronal phenotypes requires directed differentiation experiments using defined culture media. This approach aims to support the replacement of specific neural cell types lost during neurological diseases, contributing to the development of novel therapeutic strategies [24]. As mentioned above, DA neurons play a crucial role in neurodegenerative diseases such as PD. While some studies have reported neuronal differentiation from OSE and cortical-derived SCs [14,16,22], their capacity to generate DA neurons remains unexplored. Therefore, it is necessary to verify whether DA neurons can be derived from adult ovarian cortical cells (OCC), relying on previously established directed differentiation protocols with defined culture media.

Defined media for DA neuron differentiation typically contain Sonic Hedgehog (SHH) [25], fibroblast growth factor 8 (FGF8) [26], and fibroblast growth factor 2 (FGF2) [27], supplemented during the terminal differentiation phase with brain-derived neurotrophic factor (BDNF) [27,28,29]. The resulting DA neurons and their precursors express tyrosine hydroxylase (TH) as a specific marker [30], which catalyzes the conversion of L-tyrosine into L-3-4-dihydroxyphenylalanine (L-DOPA), a limiting step in the synthesis of dopamine and other catecholamines.

The NS assay provides a suitable method for assessing the self-renewal and differentiation capacity of NS-derived cells into neurons and glia [31], allowing directed differentiation experiments to generate specific neural cell populations of interest [32,33]. To further validate these findings, it would be necessary to evaluate the functionality of differentiated cells using electrophysiological techniques such as patch-clamp recording, which allows for the measurement of electrical activity in individual cells in response to stimulation [34,35]. The patch-clamp recording provides critical insight into the functional maturation and potential disease-related abnormalities of neurons. This approach is essential to ensure the functional identity and maturity of differentiated neurons for their potential use in cell replacement therapies and drug discovery platforms. The patch-clamp technique provides high-resolution data of ion channel activity, action potential dynamics, synaptic currents, and membrane properties, allowing us to assess the functional consequences of PD-related changes at the cellular level. This provides insights for understanding the complex pathophysiology of the disease and supports the development of effective therapeutic strategies and biomarkers [36].

On the other hand, assessing the survival of differentiated DA neurons is crucial for evaluating the efficacy of SC-based therapies. A recent systematic review reported that graft survival rates in preclinical PD models vary widely, with a mean survival of 51% of transplanted cells [37].

A key focus in current research on NSPCs involves identifying different signaling pathways and proneurogenic or neuroprotective biomolecules that could be used as therapeutic targets [38,39]. Recent studies have shown that follicle-stimulating hormone (FSH), secreted by the pituitary gland and involved in folliculogenesis, promotes self-renewal, clonal expansion, and differentiation of OSE-derived SCs through its interaction with the FSH receptor [21].

Moreover, FSH exerts proneurogenic effects on OCC-NS, inducing NSPCs proliferation, NS development, and the upregulation of *Nestin*, *Pax6*, and *DCX* during the NS assay [16]. As a step forward, it would be of interest to elucidate whether FSH could also enhance the differentiation of OCC-NSPCs into DA neurons during the NS assay, using previously established defined media for directed differentiation.

Therefore, the aim of this study was to generate DA neurons from OCC-NS during the NS assay by incorporating a defined differentiation medium specifically designed for this purpose. Additionally, we aimed to evaluate whether FSH could enhance this process. Finally, the functionality of differentiated neural cells obtained at the end of the culture will be assessed through electrophysiological recordings using the patch-clamp technique.

## 2. Materials and Methods

### 2.1. Cell Isolation for Culture

Ovaries from lambs (*Ovis aries*) aged 3–6 months were obtained from a slaughterhouse near the laboratory, aseptically dissected, immediately transported at 2–8 °C, and processed as previously described [14,15,16]. Briefly, 0.5–1 mm pieces of ovarian cortical tissue devoid of macroscopically visible antral follicles were dissected and subjected to enzymatic digestion using collagenase, followed by mechanical disaggregation. The resulting suspension was centrifuged, and the supernatant was replaced with a complete culture medium composed of Dulbecco’s modified Eagle’s medium (DMEM: F12; Thermo Fisher Scientific, ref. 11039-021, Grand Island, NY, USA) containing 0.1% bovine serum albumin (BSA; Sigma-Aldrich, ref. A9418, Saint-Louis, MO, USA), 3mM L-glutamine (Sigma-Aldrich, ref. G7513), 10 µL/mL insulin, transferrin, selenium (ITS; Sigma-Aldrich, ref. I3146), 2 µL/mL Synthecol (Sigma-Aldrich, ref. S5442), and 10 µL/mL antibiotic–antimycotic (Thermo Fisher Scientific, ref. 15240). This medium was successively filtered through 100, 70, and 40 μm cell strainers (Beckton Dickinson, BD Falcon, ref. 352360, ref. 352350, and ref. 352340, respectively, Brookings, SD, USA). Finally, Trypan Blue staining (Sigma-Aldrich, ref. T8154) of an aliquot of the cell suspension was performed to evaluate cell viability.

### 2.2. Experimental Design

A NS assay based on previously established procedures [14,40] was performed to assess the self-renewal and differentiation potential of NS-derived cells. This assay comprised two subsequent 7-day culture periods for cell expansion (CEP-1 and CEP-2) to obtain primary (NS1) and secondary (NS2) neurospheres, respectively. During both CEP-1 and CEP-2, image analysis and morphometric evaluation were performed every 48 h. On day 6 of each CEP, a cell proliferation test was carried out, and on day 7, RNA was extracted for analysis of NSPC-specific transcripts expression. At the end of CEP-2, NS2 were enzymatically disaggregated, and cells were cultured for 30 days under directed differentiation conditions with a defined medium for DA differentiation. During this culture for differentiation (CDP), image analysis was performed every 48 h, the expression of specific transcripts of glia, mature neurons, DA precursors, and neurons were analyzed in RNA extracts, and the corresponding antigens were immunolocalized at the end of culture. Patch-clamp electrophysiological recordings were performed on differentiated neurons to evaluate their functionality at the end of the culture.

### 2.3. Culture for Cell Expansion

After cell isolation, 500,000 and 20,000 live cells were seeded per well in 24-well (Nunclon Delta; Nalgene Nunc, ref. 142475, Rochester, NY, USA) and 96-well (Nunclon Delta; Nalgene Nunc, ref. 167008) plates, respectively. These plates had been pre-coated with 1 µg/mL fibronectin solution (Sigma-Aldrich, ref. F4759) [14]. Cells were maintained for 7 days in both CEP-1 and CEP-2 at 37 °C, 5% CO_2_, and 99% humidity in a Forma Steri-Cycle incubator (Thermo Scientific Forma, Stericycle Sensor TC; Thermo Fisher Scientific, Lima, OH, USA), with one of the following three defined culture media:

M1, consisting of DMEM: F12, 0.1% bovine serum albumin, 3 mM L-glutamine, 1% N2 supplement (Thermo Fisher Scientific, ref. 17502048), 20 ng/mL epidermal growth factor (EGF; Sigma-Aldrich, ref. E9644), 20 ng/mL fibroblast growth factor 2 (FGF2; Sigma-Aldrich, ref. F0291), and 10 µL/mL antibiotic–antimycotic. This medium was used as a positive control for cell expansion and neural induction [41].

M2, composed by DMEM: F12, 0.1% BSA, 10 µL/mL insulin–transferrin–selenium supplement, 3 mM L-glutamine, 2 µL/mL of Synthecol and 10 µL/mL of antibiotic–antimycotic. This medium was used as the control.

M3, with the same composition as M2, and supplemented with 50 ng/mL ovine FSH (o-FSH; National Institute of Diabetes and Digestive and Kidney Diseases, ref. NIDDK-oFSH-18AFP5862D, Bethesda, MD, USA).

The culture medium was replaced every 48 h with a fresh medium with the same composition. At the end of CEP-1 and CEP-2, NS1 and NS2 were enzymatically disaggregated using Accutase (StemPro Accutase; Gibco, Thermo Fisher Scientific, ref. A11105-01) following the manufacturer’s recommended protocol and subcultured to initiate CEP-2 or directed CDP, respectively.

### 2.4. Culture for Directed Differentiation Towards DA Pathway

After disaggregating NS2 and performing the Trypan Blue viability test, cells were seeded in 24-well plates at a density of 86,000 cells per cm^2^ (144,000 cells per well) on 12 mm diameter round borosilicate coverslips (Menzel, Thermo Fisher Scientific, ref. CB00120RAC20MNTO), pre-coated with poly-L-ornithine (Sigma-Aldrich, ref. P4957) and fibronectin (1 µg/mL per well; Sigma-Aldrich, ref. F4759) according to a standardized protocol (Protocol for Coating Coverslips with Poly-L-ornithine and Fibronectin; R&D Systems Protocols, Minneapolis, MN, USA). Cells were maintained at 37 °C, 5% CO_2_, and 99% humidity for 30 days, as previously described [14], in a defined culture medium for directed DA neuron differentiation [18] with some modifications. This medium contained Neurobasal medium (Thermo Fisher Scientific, ref. 12348-017), 0.1% BSA, 2mM L-glutamine, 10 µL/mL non-essential amino acid solution (NEAA, Gibco, Thermo Fisher Scientific, ref. 11140050), 1% N2 supplement, 250 ng/mL Sonic Hedgehog (SHH, R&D Systems, ref. 1845-SH-025), 200 ng/mL FGF8 (R&D Systems, ref. 423-F8), 100 ng/mL FGF2, and 10 µL/mL antibiotic–antimycotic. From day 9 of culture for differentiation onwards, 50 ng/mL of brain-derived nerve factor (BDNF; Invitrogen, Life Technologies, ref. 10908010) was added to this defined medium.

### 2.5. Image Analysis and Morphometric Evaluation

Cultures were observed and photographed under an inverted microscope (Nikon Eclipse T*i*S; Nikon Instruments Inc., Melville, NY, USA) equipped with a digital camera (Nikon DS-Fi1; Nikon Instruments Inc.) and image analysis software (NIS-D-Elements, Nikon). Morphometric evaluation was carried out by measuring the diameters of at least 150 NS at each time point, as previously described [14], with the formation of NS1 and NS2 serving as the main indicator of their self-renewal capacity.

### 2.6. Cell Proliferation Assay

Cell proliferation was assessed in 96-well plates on day 6 of culture in both CEP-1 and CEP-2, using a commercial colorimetric analysis kit (BrdU Cell Proliferation ELISA kit; Abcam, ref. ab126556, Cambridge, UK) following the manufacturer’s instructions. Bromodeoxyuridine (BrdU) uptake into the cellular DNA in M1, M2, and M3 cultures (*n* = 7 culture wells/group/time point) was recorded at 450/630 nm in a Stat Fax 2100 ELISA microplate reader.

### 2.7. Gene Expression Analysis

Total RNA was extracted from NS1 and NS2 lysates obtained on day 7 of each CEP and on day 30 of the directed CDP, using six wells per group. RNA samples were stored at −80 °C until processed at the Antonia Martín Gallardo Genomics and Proteomics Service, Madrid Science Park, Cantoblanco Campus, as previously described [14]. During the CEPs, transcripts characteristic of NSPCs were quantified by quantitative Reverse Transcription Polymerase Chain Reaction (qRT-PCR), including sex-determining region Y-box 2 (*Sox2*), *nestin*, paired box 6 (*Pax6*), and neurotrophin receptor p75 (*p75NTR*). At the end of the directed CDP, the expression of differentiation-related transcripts was analyzed, including glial fibrillary acidic protein (*GFAP*), a characteristic astrocyte/radial glia transcript; oligodendrocyte-specific transcript (*Olig2*); neuronal nuclei (NeuN), a biomarker for differentiated neurons; and TH, a marker for DA precursors and neurons. 18S ribosomal RNA was used as an endogenous control. For each gene, the expression of each transcript in cells from group M1 or M2 was used as a reference for relative quantification (RQ) in the CEPs or the directed CDP, respectively. Primers used for amplification are listed in Table 1 and were synthesized based on mRNA sequences available at the National Center for Biotechnology Information.

### 2.8. Immunolocalization of Antigens Characteristic of Neural Differentiation

Immunofluorescent staining for the neural differentiation antigens GFAP, NeuN, and TH was assessed on day 30 of the directed CDP following a previously established protocol [10]. Cells were incubated with fluorochrome-conjugated antibodies diluted in 5% BSA-PBS. Anti-NeuN (1:300; Abcam, ref. ab177487) was labelled with DyLight 650 (Abcam, ref. ab201803); anti-GFAP (1:200; Sigma-Aldrich, ref. G4546) with DyLight 594 (Abcam, ref. ab201801); and anti-TH (1:200; Thermo Fisher Scientific, ref. PA5-85167) with DyLight 488 (Abcam, ref. ab201799). An Olig2 antibody with reactivity in sheep was not available. Individual (NeuN, GFAP, or TH) or combined (NeuN + TH; NeuN + GFAP) preparations were performed in each group, along with negative control cell replicates in which incubation with blocking solution replaced that of the primary antibody. After overnight incubation with antibodies, glass slides were washed with cold PBS, counterstained, mounted with DAPI Prolong Gold, and stored in the dark at 4–8 °C until analyzed under a confocal laser microscope (Leica TCS SP8; Leica Microsystems, Wetzlar, Germany) at the Center of Cytometry and Fluorescence Microscopy, of Complutense University of Madrid.

To analyze the TH staining area in each confocal plane and its ratio with respect to the total area of the corresponding plane, RGB images of immunostained NS for each individual channel (green for TH; blue for DAPI) were saved as uncompressed TIFF files for analysis using image analysis software, (Fiji ImageJ 1.54p, National Institute of Health, Bethesda, MD, USA). For TH analysis, the RGB images were used to obtain the 8-bit grayscale image of the green channel. The following parameters were measured: the TH staining area, the fraction of the TH-staining area, minimum and maximum grey values, mean grey value, standard deviation, and median. TH staining data were normalized by subtracting background staining values of negative controls. Processing of the TH-stained section series was automated using a macro comprising conversion of images to grayscale, applying a Gaussian Blur filter (sigma: 1), automatic thresholding (Moment’s method), and setting measurement parameters.

The total area of each confocal plane was calculated using TIFF files obtained from DAPI staining (blue channel) by manually outlining the complete plane contour. The TH-staining area of each confocal plane was then calculated as the ratio between the normalized TH-staining and the total area of the corresponding plane. Summatory of TH-staining areas (ST), summatory of total plane areas (SA), and ratio ST/SA were calculated for each NS.

### 2.9. Electrophysiological Study: Patch-Clamp Technique

Patch-clamp recordings were carried out on day 30 of culture during the directed CDP following previously established protocols [34,35,42,43]. Coverslips from each of the three experimental groups were selected and placed in a recording chamber where cells were exposed to an external ionic solution containing 125 mM sodium chloride, 1 mM magnesium chloride hexahydrate, 3 mM calcium chloride dihydrate, 2.5 mM potassium chloride, 10 mM HEPES and 20 mM glucose at pH = 7.3, 280–290 mOsm/L with a flow rate of 1 mL/min at 25 °C, using a temperature controller (TC-324C Warner Instruments). Cells were visualized under a 40X water immersion objective and a Nomarski condenser combined with infrared microscopy using differential interface contrast (DIC) on an Eclipse FN1 Nikon microscope. Electrophysiological recordings from selected cells (*n* = 15) were obtained using a PC-ONE amplifier and patch pipettes (3–4 MΩ resistance) pulled from thick-walled borosilicate glass (1.5 mm outer diameter and 1.1 mm inner diameter) on a P-97 puller (Sutter-Instrument) and filled with an internal solution containing: 135 mM potassium gluconate, 10 mM potassium chloride, HEPES 10 mM, 1 mM magnesium chloride, and 2 mM adenosine triphosphate disodium salt, at pH = 7.3, and 280–290 mOsm/L. After the formation of a stable whole-cell configuration (–70 mV holding potential), neutralization of fast and slow whole-cell capacitances, and compensation of series resistance (70%), cells were transferred to current-clamp mode and injected with a series of increasing currents (20-pA step, 500-ms duration with a 3-s interval) to analyze the action potential firing pattern. Signals were fed to a Pentium-based PC through a DigiData1322A interface board (Axon Instruments), and the pCLAMP 10.2 software was used to generate stimuli, as well as for data display, acquisition, storage, and analysis.

### 2.10. Statistical Analyses

One-way analysis of variance (ANOVA) was performed to assess the influence of experimental groups (M1, M2, and M3) and time on cell proliferation (absorbance values) and NS development in culture (diameters). Differences in data from RQ of transcripts analyzed in CEPs and CPD were determined by one-way ANOVA, except for data from Olig 2 and TH that were analyzed by Kruskal Wallis Non-parametric test because data did not comply with a Gaussian distribution. In all cases, the Bonferroni multiple-comparison test was used as an ANOVA post-hoc test, and the Dunn multiple-comparison test was used as the Kruskal–Wallis post-hoc test. The effect of the experimental group (M1, M2, M3) on the ratios ST/SA was determined by Unpaired-T test analysis. Results are presented as percentages with standard error. Differences were considered significant with *p* < 0.05, *p* < 0.01, and *p* < 0.001. Statistical analyses were performed by using SAS software (Statistical Analysis System 9.4, SAS Institute Inc., NC, USA; Informatic Services Complutense University of Madrid) and Graph Pad Prism 4 software (GraphPad Software Inc., CA, USA).

## 3. Results

### 3.1. NS Assay: CEPS

#### 3.1.1. Image Analysis and Morphometric Evaluation of NS in Culture

Microscopy observations revealed the spontaneous formation of cell aggregates 72 h after the onset of culture in both CEPs, which subsequently developed into NS by days 5–7 of culture (Figure 1A,B). Mitosis was frequently observed in both CEPs (Figure 1C). Cells with an elongated morphology and neural-like appearance were located at the periphery of the NS and adhered to the growth surface. In the early stages, these cells exhibited a central nucleus, abundant cytoplasmic lipid-like droplets, and numerous plasma membrane protrusions, such as filopodia, which later transformed into neurite-like projections in culture (Figure 1D). As the culture progressed, some of these membrane projections elongated (Figure 1E), eventually acquiring the characteristic morphology of neurons (Figure 1F).

NS diameters increased in all groups after seeding in both CEPs (Figure 2A,B). In CEP-1 (Figure 2A), NS1 diameters (μm) in M1 were larger (*p* < 0.05) on day 5 compared with M2 (77.70 ± 2.97 vs. 66.62 ± 2.55, respectively). By day 7, NS1 diameters in M1 (106.40 ± 5.12) and M3 (119.30 ± 5.81) were larger (*p* < 0.01) than those in M2 (87.19 ± 3.35). In CEP-2 (Figure 2B), NS diameters reached their maximum size on day 7 in all groups. NS2 diameters were larger in M1 (139.00 ± 7.25, *p* < 0.05) and M3 (144.60 ± 11.78, *p* < 0.01) compared with M2 (105.80 ± 6.07).

#### 3.1.2. Cell Proliferation Assay

Cell proliferation occurred in cells cultured in M1, M2, and M3 during both CEPs. In CEP-1, BrdU incorporation was greater in cells from M1 (0.82 ± 0.04) and M3 (0.91 ± 0.07) than in those from group M2 (0.32± 0.13, *p* < 0.01 and *p* < 0.001, respectively). In CEP-2, M3 cells exhibited the highest proliferative activity (1.34 ± 0.11), greater than M2 (0.92 ± 0.06, *p* < 0.01) and M1 (0.16 ± 0.04, *p* < 0.001), with M2 cells also showing higher proliferation than M1 cells (*p*< 0.001). Image analysis indicated that cells exposed to EGF and FGF2 (M1) in CEP-2 reached confluence by day 3 of culture.

#### 3.1.3. Gene Expression Analyses

During CEP-1 (NS1), expression of NSC/NPC-specific transcripts *Sox2*, *nestin*, *Pax6* and *p75NTR* was higher in M2 than in M1 and M3 (Figure 3A–D). *Sox2* expression was greater (*p* < 0.01) in M2 cells than in M1 (Figure 3A), as was *Nestin* in M2 (*p* < 0.05) when compared with M3 (Figure 3B). The transcription of *Pax6* and *p75NTR* (Figure 3C,D) was greater in M2 cells than in M1 (*p* < 0.01) and M3 (*p* < 0.05) cells. During CEP-2 (NS2), *nestin* expression was similar (*p* > 0.05) in cells from all three groups (Figure 3E), *Sox2* expression increased (*p* < 0.05) in M2 cells when compared with that of M1 cells (Figure 3F), and transcription of *Pax6* in M2 cells increased over values of M1 (*p* < 0.001) and M3 (*p* < 0.01) cells (Figure 3G). Expression of *p75NTR* was higher in M2 (*p* < 0.01) and M3 (*p* < 0.05) cells compared to that of M1 (Figure 3G,H) cells.

### 3.2. NS Assay: Directed CDP

#### 3.2.1. Image Analysis

During this stage of culture, a significant proportion of cells adhered to the growth surface and gradually changed their morphology from a round shape with a high nucleus-to-cytoplasm ratio to the emission of filopodia from the plasma membrane and the development of neurite-like membrane projections (Figure 4A). These cells often displayed abundant lipid droplets in their cytoplasm (Figure 4B). Both cytoplasmic and membrane projections progressively elongated (Figure 4C), and images consistent with early stages of neurogenesis, such as developing neuroblasts and radial glia, were also observed (Figure 4D). Eventually, these projections were progressively transformed into fewer structures resembling axons or projections of variable length and greater abundance in specific regions of the cell surface, like dendrites (Figure 4E,F). Additionally, cells with characteristic astrocytic morphology were frequently observed. During this period, spontaneous NS formation remained as a residual phenomenon, particularly evidenced in cells cultured in the M3 group during CEPs, and occurred less frequently in cells cultured in M1 during CEPs. NS generated during differentiation were consistently located at the periphery of the growth surface, separated from differentiating cells. Differentiating cells cultured in the M1 group during CEPs exhibited more defined neuronal differentiation morphologies during CDP.

#### 3.2.2. Gene Expression Analyses

The expression of transcripts characteristic of differentiated neurons (*NeuN*), astrocytes (*GFAP*), oligodendrocytes (*Olig2*), and DA neuronal precursors/DA neurons (*TH*) during CDP are presented in Figure 5. Cells from the M2 group exhibited the lowest levels of expression of all transcripts analyzed, which were used as references for the RQ of the studied genes in cells from the M2 and M3 groups. Transcriptions of *NeuN* and *GFAP* (Figure 5A,B) were higher in differentiating cells from groups M1 and M3 during the CDP compared to those from M2 (*p* < 0.001 for *NeuN* in both cases; *p* < 0.01 and *p* < 0.001, respectively, for *GFAP*). The expressions of *Olig2* and *TH* (Figure 5C,D) in cells from M1 were greater than those in cells from M2 (*p* < 0.01 and *p* < 0.05, respectively).

#### 3.2.3. Immunolocalization of Neural Cell-Specific Antigens

A large proportion of cells that adhered to the growth surface and exhibited morphological features characteristic of neural differentiation showed intense immunolocalization of NeuN, a marker of differentiated neurons (Figure 6A,B; Appendix A). NeuN immunolocalization was also observed in migratory or post-migratory neuronal precursor cells, where occasional colocalization with TH, an enzyme involved in DA synthesis in neurons and DA neuron precursors, was also observed (Figure 6D; Appendix A). NeuN immunolocalization was consistently found in cells from the peripheral strata of the NS, with high staining intensity (Figure 6E–I, Figure 7 and Figure 8). Both NeuN immunolocalization and staining intensity decreased as the cells approached the NS-core from the outer cell layer. In fact, NeuN immunolocalization was absent in the inner cell core (Figure 6E,G–I). TH immunolocalization was detected in neuronal precursors found outside the NS (Figure 6D), exclusively in the “core” cells of small (Figure 6E) and medium-sized NS (Figure 6G–I). Large NS, more frequently found in groups M2 and M3, contained cells with positive immunolocalization to TH both in the core and entering the outer cell layer (Figure 7 and Figure 8; Appendix A). GFAP labelling was moderately frequent among cells growing on the seeding surface (Figure 6C; Appendix A).

The mean ST/SA ratio, expressed as the percentage of plane areas immunolocalizing TH was affected by cell exposure to M1, M2, or M3 during CEP1 and CEP2. The highest percentage of TH-immunolocalized areas was observed in NS cultured with M3 medium (with FSH) during CEP1 and CEP2 (33.50 ± 4.9), which was higher (*p* < 0.05) than in NS from M2 (23.74 ± 2.3) and M1 (29.35 ± 2.0).

#### 3.2.4. Electrophysiological Study: Recording of Electrical Activity Using the Patch-Clamp Technique

To characterize the degree of functional maturation of CDP-directed cultures, cells were selected by optical microscopy and subjected to patch-clamp recordings (Figure 9A). In current-clamp mode, we analyzed the cellular response to a series of increasing current injections (Figure 9B–D) to assess the cell’s ability to fire action potentials and the firing pattern. Three different patterns were observed in these recordings. First, electrophysiological pattern 1 (*plateau* potential) was detected in a significantly higher number of recordings (11 of 15 cells). In this case, the cells responded to increasing depolarizing currents with a single action potential (Figure 9B). Second, electrophysiological pattern 2 (passive electrical response) was detected in a small number of recordings (2 of 15) in which the cell did not actively respond to injected depolarizing currents with membrane depolarization (Figure 9C). Third, electrophysiological pattern 3 (action potential) was also detected in a small number of recordings (2 of 15), where the cell responded to increasing depolarizing currents with a proper action potential, exhibiting the characteristic depolarization spike followed by membrane repolarization (Figure 9D).

## 4. Discussion

This study demonstrates, for the first time, that NSPCs derived from OCC-NS can differentiate into DA neuron precursors and neurons under previously established directed differentiation conditions [14]. Moreover, the addition of FSH to the defined culture medium during the CEPs promotes neurogenic development, increasing the expression of transcripts characteristic of mature neural cells, including TH expression and immunolocalization of its corresponding protein. It also provides the first experimental evidence of physiologically relevant electrical cell responses in neural cells differentiated from OCC-NSPCs.

This is the first research in which FSH priming during CEPs of the NS assay is combined with directed differentiation of NSPCs towards the DA pathway by using a defined medium containing SHH, FGF8, and BDNF, a procedure with possible eventual clinical implications for Parkinson’s disease therapy.

The results of this study confirm previous contributions regarding the proneurogenic actions of FSH on OCC-NSPCs (16) by demonstrating again that this hormone increases NSPCs proliferation and self-renewal. The results also provide further progress in defining these actions by showing that priming NSPCs with FSH during CEPs upregulates the expression of transcripts specifically associated with terminal neuronal differentiation, *NeuN*, and, remarkably, *TH* (transcript and protein) after directing the differentiation of these cells towards the DA pathway. Our results show that FSH (M3) upregulates the mean *NeuN* expression between two to three times with respect to control (M2), similar to the positive control group (M1). *TH* transcription of differentiated cells primed with FSH during CEPs (M3) increases up to three to four times the expression quantified in control (M2), with values slightly larger but non statistically significant with respect to the positive control group (M1). Outstandingly, the actions of FSH on *TH* transcription during differentiation are neatly reflected in the percentages of NS cells exhibiting TH immunolocalization, with the largest percentages quantified after DA differentiation of NSPCs primed with FSH in CEPs (33.5%) over values of control (23.74%; M2), and not significantly larger than the positive control group (M1; 29.35%; M1). Several other aspects of these results will be further discussed at the end of this section.

Results showed that NS cells proliferate, self-renew, and express transcripts characteristic of NSPCs throughout both CEPs, as previously described [14,15,16]. Cell aggregation was observed in all three experimental groups from day 3 of culture onwards, with a progressive increase in NS diameters. These diameters were greater in groups M1 and M3 when compared with group M2 at the end of CEP-1. Group-dependent NS development was consistent with the greater proliferative activity observed in cells from groups M1 and M3. This supports, on the one hand, the role of EGF and FGF2 as inducers of cell proliferation and aggregation to generate spheroids [44] and, on the other, the proneurogenic actions of FSH, which stimulates proliferation [45,46], self-renewal, and increases NS size [16]. These actions are likely enhanced by the paracrine interaction between SC and stromal cells from the ovarian cortical tissue, which secrete neural-inducing factors in a similar fashion to the system known as stromal cell-derived inducing activity (SDIA) [47]. EGF and FGF2 are neural-inducing factors secreted in brain neurogenic niches that reactivate quiescent neural stem cells, promoting their proliferation and differentiation into neurons and glial cells to replace damaged cells. EGF and FGF2 exert these same actions as part of normal neurogenesis [48,49]. These effects are clearly reproduced in the experiments conducted in the current research, which is an indication that OCC-NSC should be further explored for their potential utility in autologous cell therapy. High proliferation, self-renewal, and differentiation abilities make SCs suitable for regenerative medicine and cell therapy [50]. However, as a possible side-effect, the tumor formation potential of these cells must be thoroughly investigated as part of the safety analyses, provided that EGF and FGF2 are also involved in tumor progression due to their proliferative actions on cancer cells [51].

During CEP-2, the results also showed reduced BrdU incorporation on day 7 in group M2 compared with groups M1 and M3.

The addition of FSH to defined culture medium during CEPs (M3) upregulated the expression of transcripts characteristic of NSPCs on day 7 of culture, like that observed when cells were exposed to neural induction factors (M1). These results support the hypothesis that FSH added to the culture medium might stimulate the synthesis of EGF and FGF2 by SCs [52,53], which would act in an autocrine/paracrine fashion to induce cell proliferation and prevent the onset of differentiation [54]. This could also explain the lower expression of *Pax6* and *p75NTR* in M1 and M3 cells during the first days of culture when compared with their transcription in M2 cells. Further evidence supporting this hypothesis is the greater transcription of the neural progenitor marker *Pax6* [55] and the lower proliferative activity in cells from group M2. In relation to the expression of *Pax6*, recent studies have shown that it increases between days 15 and 21 of culture during the differentiation of OCC-NS cells into mature neural cells [14,16]. Taken together, these results suggest that FSH (M3) mimics the proliferative and neural-inducing effects of EGF and FGF2 (M1), thereby maintaining NSC self-renewal during CEPs. In contrast, NSPCs cultured without these factors or FSH (M2) exhibit reduced self-renewal and initiate differentiation.

During directed CDP, NS cells differentiated into neurons and glial cells, supporting the identity of the spheroids as NS, as previously described [14]. It is noteworthy that the composition of the culture medium during both CEPs significantly influenced the expression of transcripts characteristic of differentiated neural cells (*NeuN*, *GFAP*, *TH*), which were generally higher in M1 and M3 compared with M2. This suggests that EGF, FGF2, and FSH may exert similar proneurogenic actions.

Previous studies have demonstrated the promoting actions on neurogenesis of several hormones, such as prolactin, oxytocin, and estrogens [56], GnRH [57], and LH [58]. Recent research has shown that NSPCs primed with FSH during CEPs exhibit increased expression of the *DCX* neuron precursor transcript [16]. FSH receptor (FSHr) is expressed in the cerebral cortex and hippocampus, and its blockade has been shown to slow the progression of Alzheimer’s disease in women [59].

The protocol proposed by Faghih et al. [18], used to generate DA neurons during the directed CDP, successfully allowed DA precursors to be obtained in the current experiments. Recently, other authors have also derived DA neurons from testicular germ SCs [20].

A notable finding during directed CPD was the discrete and residual generation of NS, although in smaller quantities compared with those observed during CEPs. During the differentiation period, NSPCs gradually lose their proliferative capacity. However, there appears to be a constitutive synthesis and secretion of EGF and FGF2 [60], which maintains basal NS formation. In fact, the addition of EGF and FGF2 antagonists is compulsory to block their proliferative actions and drive NSPCs towards differentiation [14]; however, these antagonists were not incorporated in the defined medium of directed CPD in the current experiments because the actions of FGF2 and FGF8 are specifically required to drive the cells to the DA pathway [26,49,61].

The presence of neurons, astrocytes, and oligodendrocytes was confirmed by gene expression of *NeuN*, *GFAP*, *Olig2*, and *TH*, as well as by positive immunolocalization of NeuN, TH, and GFAP, confirming the presence of differentiated neurons, glial cells [14], and DA neuron precursors.

*TH* gene expression and its positive immunolocalization indicate that some cells present at the end of the directed CDP are DA neurons or DA neuron precursors since TH was immunolocalized in a proportion of cells with a characteristic neuronal morphology adhered to the growth surface in big round neural precursor cells, in both cases showing colocalization with NeuN (NeuN^+^/TH^+^, DA neuron precursors). An interesting finding was the consistent and abundant immunolocalization of TH (NeuN^−^/TH^+^; DA neuron precursors) in core cells of NS that residually continue their generation during CDP. TH was immunolocalized in core cells of all NS regardless of size and experimental group and in cells from the core entering the outer cell sheet of the NS, found in large NS frequently found in groups M2 and M3. Remarkably, image analyses of large NS suggest that DA neuron precursors apparently invade the outer cell sheet to presumably abandon the NS, giving rise to migratory DA neuron precursors ready for terminal differentiation outside the NS on the growth surface. This hypothetical explanation, supported by image analyses in this study, will be addressed in future experiments.

It is noteworthy that the location of the DA neuron precursors in this study coincides with findings in NS derived from midbrain neuronal precursors [62]. The localization of DA precursors in the core of the NS likely occurs because this area contains NSPCs that have not yet initiated differentiation during the directed CDP and, therefore, respond to factors that direct their fate towards the DA pathway.

Future studies should incorporate adjustments to the NS assay, such as isolating and culturing only NSPCs from NS generated during the directed CDP, to produce a more homogeneous and plentiful population of DA neurons. Several authors have noted that DA neuron generation from NS depends on several factors, such as regional specificity, donor tissue age, duration of in vitro culture, and culture conditions [63]. However, attempts to generate a larger number of DA neurons from NSCs originally obtained from the embryonic CNS have not yielded large proportions [64,65]. Some studies have indicated that the percentage of TH^+^ cells is less than 2% of the total population in NSC derived from the embryonic rat midbrain, even when genetically modified to express transcription factors associated with DA differentiation [66]. In contrast, TH expression in neurons differentiated from hair follicle-associated pluripotent SCs cultured in a neural induction medium reached 48.90 ± 4.64% [67], while that of neurons differentiated from postnatal hippocampal NSC reached 11.9 ± 1.68% on day 7 under low-serum conditions [68]. As reference values, TH expression in differentiated midbrain DA neurons ranges from 20% to 75%, depending on the induction protocol used [69]. Despite possible methodological differences, our results show immunolocalization percentages between 23% and 33% relative to the total area analyzed in NS.

Furthermore, future studies should also address the selective generation of the DA neuron phenotype from this assay. These neurons should exhibit characteristics specific to the *substantia nigra pars compacta* to ensure an adequate integration in the damaged area for clinical applications [70,71].

It is of interest to explore, in this culture system, the reliability of analyzing the expression and localization of other enzymes involved in catecholamine biosynthesis, such as aromatic L-amino acid decarboxylase and DA β-hydroxylase. Recent studies suggest that dysregulation of these enzymes contributes to the selective vulnerability of DA neurons, which may serve as complementary therapeutic targets in PD. This would also aid in generating strategies that modulate multiple enzymes in the DA pathway, contributing to a greater understanding of the mechanisms involved in PD pathogenesis [72].

A large proportion of morphologically differentiated cells covering the growth surface during directed CPD were NeuN^+^/TH^−^. One possible explanation of this finding is that Accutase dissociation was performed on all cells present in the culture at the end of CEP-2, including both NS and cells attached to the growth surface that appeared to have initiated differentiation based on morphological criteria. These latter cells might have already been committed to a specific fate before the onset of directed CDP, whereas cells that transitioned from CEP-2 to directed CDP as NSPCs were able to respond to the factors present in this differentiation medium, directing them towards a final DA fate.

A singular contribution of this study refers to the inductive actions of FSH (group M3) during CEP on the expression and synthesis of the enzyme TH, localized as mentioned above. This is most likely the result of the proneurogenic actions of this hormone on OCC-NSPCs, such as the upregulation of *nestin*, *Sox2*, *Pax6*, and *DCX* [16]. Consequently, NSPCs from group M3 exhibit greater neurogenic potential when exposed to DA-driving factors (FGF2, FGF8, and SHH), which would explain their increased efficiency in terminal differentiation into DA neuron precursors. This is supported by the greater TH expression and synthesis in this group. On the other hand, no studies have reported a direct relationship between FSH and *TH* expression or synthesis. However, several authors have pointed out that some steroid hormones, such as estradiol and progesterone [73,74,75], could influence TH expression, with prolactin and possibly other pituitary hormones involved in this mechanism [76].

To our knowledge, this is the first study to analyze the electrophysiological responses of neurons differentiated from OCCs in culture using the patch-clamp technique. This contribution offers the possibility of setting up more complex studies related to altered ion channel function as underlying or associated pathophysiological mechanisms in neurodegenerative diseases using this model. Studies on membrane ion channels provide promising therapeutic insights for neurodegenerative diseases. Ion channels play a central role in regulating neuronal excitability, synaptic activity, and intracellular calcium signaling, processes that are often dysregulated in diseases such as Parkinson’s [77,78]. Therefore, evaluating the electrophysiological properties of differentiated neurons, for instance, through patch-clamp recordings, provides essential information on their functional identity and therapeutic potential [79,80]. The results revealed three types of characteristic electrical responses: most recorded cells exhibited plateau potentials, while the remaining cells showed either passive electrical responses or action potentials characteristic of mature neurons. The latter two types of responses have been previously described in patch-clamp studies involving neurons differentiated from humans [81] and murine embryonic cells [35]. Regarding the two active electrical responses observed, plateau potentials and the proper action potentials, these are associated with the early and late stages of neural development, respectively [82]. Both types of recordings are likely due to temporal variation in the expression of components of voltage-gated calcium, sodium, and potassium ion channels during the in vitro differentiation of NPCs into neurons. This variation has been described during the neurogenesis of mesencephalic neurons, where plateau potentials mediated by calcium channels were observed during the embryonic stage, gradually being replaced by action potentials with predominant sodium and potassium conduction [83]. Similar variations have been observed during neurogenesis in frog spinal cord neurons [84] and in sphinx moth brain neurons [85].

Plateau potential precedes complete membrane repolarization and appears in the recording when there are few voltage-gated potassium channels, and the cell possesses voltage-dependent calcium channels, which are characteristic of immature cell stages. In line with these results, plateau potentials have been described in immature neurons derived from human embryonic SCs in culture, where there is low sodium conductance [83]. This is likely due to the reduced expression of proteins that form the voltage-gated sodium ion channel, which is primarily involved in membrane depolarization during the action potential. More recent studies have reported similar electrophysiological responses during the differentiation of murine embryonic SCs [35].

In our study, most of the recorded cells (13 of 15) exhibited action potentials, although with a single frequency. These results are similar to those obtained on day 22 of culture with induced pluripotent SCs derived from human fibroblasts under DA differentiation conditions. These cells were able to produce isolated action potentials, like those observed in this study, while neurons in longer-duration cultures were able to experience trains of action potentials [86].

In the current study, action potentials were recorded on day 30 under differentiation conditions. The presence of action potentials, even if isolated, indicates that these neurons have sufficient voltage-gated sodium [87] and potassium [88] channels to support membrane depolarization and repolarization, respectively, and suggests that the neurons have completed their electrophysiological development.

In contrast, passive electrical responses reflect the low-intensity current flow corresponding to the passage of the solution used in the technique into the punctured cell’s cytoplasm through the pipette. In these cases, the cell does not respond to the stimulus for two possible reasons. First, cellular damage during the technique may compromise cell viability. Second, the developmental stage may be insufficient for the cell to respond to the stimulus, as seen in immature stages of neurogenesis, such as pre-migratory neuroblasts [87]. In the current study, recordings were performed on neurons selected based on morphological criteria indicative of maturity, consistent with previous findings showing NeuN-positive immunostaining [14]. However, cells not exhibiting full morphological differentiation but selected for path-clamp due to the presence of concentrated cytoplasm and nucleus regions, as well as nerve cell projections, were also included. Although precise correspondence remains undetermined, a significant portion of the passive electrical activity recordings likely belong to this second type of cell.

## 5. Conclusions

This study establishes the experimental basis for the generation of DA neuron precursors and mature neurons from OCCs-NSPCs. It demonstrates that priming these cells with FSH during CEPs of the NS assay increases the expression of neural differentiation transcripts, similar to the neural induction factors EGF and FGF2, and particularly *TH* expression and its corresponding protein, the enzyme responsible for converting tyrosine into L-DOPA, a key step in dopamine biosynthesis. A significant proportion of the differentiated cells at the end of the culture period exhibited functional electrophysiological responses, such as plateau and action potentials, confirming their neuronal identity. Importantly, this study advances previous findings by reporting, for the first time, the combined use of FSH during NSPC expansion with a previously established directed differentiation procedure to drive these cells toward the DA pathway. Future research will aim to refine the NS assay to generate a more homogeneous and abundant population of DA neurons that ensures the selection of functionally relevant neuron phenotypes and to perform comprehensive analyses that will help to elucidate whether NSPCs can be safely used in eventual autologous cell therapy. The potential application of these cells in neurodegenerative disease modelling and autologous cell therapy trials is currently under consideration.

## Figures and Tables

**Figure 1 biomedicines-13-01560-f001:**
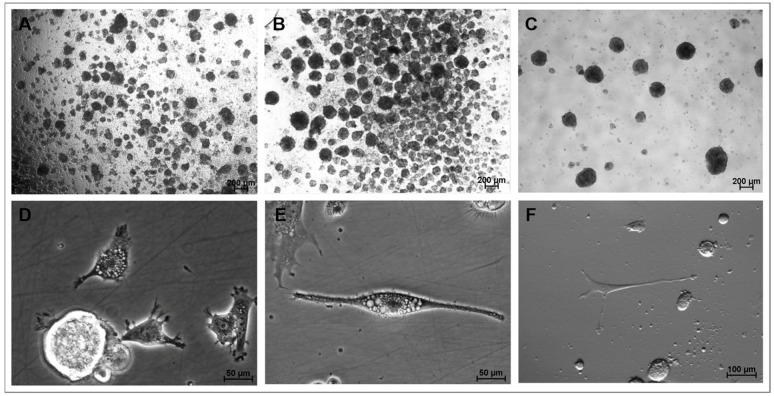
Representative photomicrographs obtained during the CEPs of the NS assay showingNS1 formation on day 5 of expansion, prior to the compaction of the aggregatess (achromatic plane, 4X/0.20 magnification) (**A**); NS1 on day 7 of culture in group M2 (achromatic plane, 4X/0.20 magnification) (**B**); NS2 on day 3 of culture in group M1 (achromatic plane, 4X/0.20 magnification) (**C**); cells on day 5 in the CEP-2 (phase contrast, 40×/0.55 magnification) (**D**); cell with fusiform morphology/neuroblast on day 7 in the CEP-2 (phase contrast, 40×/0.55 magnification) (**E**); cell with neuronal morphology on day 7 in the CEP-2 (Hoffman, 20×/0.40 magnification) (**F**). NS1 and NS2, neurosphere in the first and second cell expansion period, respectively. CEP, cell expansion period.

**Figure 2 biomedicines-13-01560-f002:**
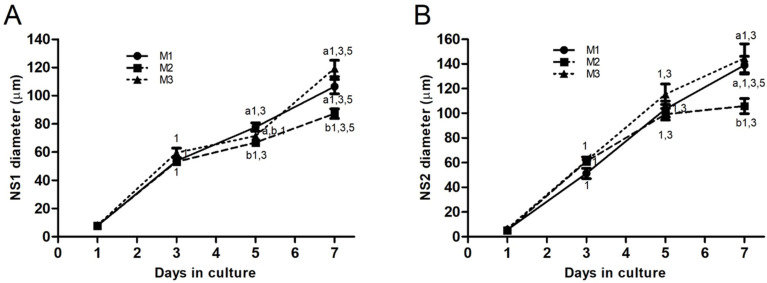
Line graphs depicting the time-course evolution of NS diameters (mean ± standard error) derived from ovarian cortical tissue cells during CEPs. (**A**) Mean diameters of NS1 during CEP-1 and (**B**) mean diameters of NS2 during CEP-2, in groups M1, M2, and M3. Different numbers indicate significant differences (*p* < 0.01) among groups within each day in culture. Different letters denote significant differences (*p* < 0.01) over time in culture within each group. NS, neurosphere; NS1, primary neurospheres; NS2, secondary neurospheres; CEP-1 and CEP-2, first and second cell expansion periods.

**Figure 3 biomedicines-13-01560-f003:**
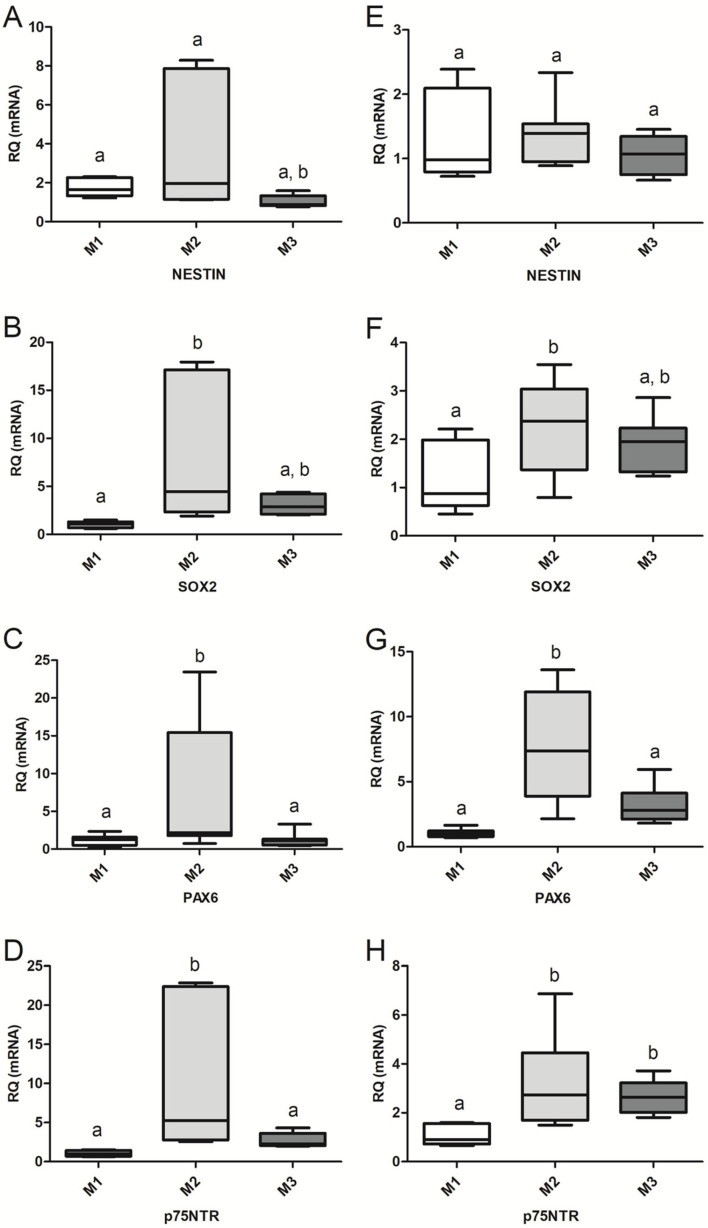
Graphic representation (boxes and whiskers) showing the expression levels of *nestin* (**A**,**E**), *Sox2* (**B**,**F**), *Pax6* (**C**,**G**), and *p75NTR* (**D**,**H**) in primary neurospheres (NS1; **A**–**D**) and secondary neurospheres (NS2; **E**–**H**). The expression of each transcript in NS from group M1 was used as the reference for the relative quantification (RQ) of each gene. The median (center line of each box) of the RQ values for each gene is represented, with the first and third quartiles of the results as the upper and lower limits of the box and the upper and lower whiskers as the maximum and minimum values, respectively. Different letters indicate significant differences among groups with *p*-values as follows: *p* < 0.001 (NS2: *Pax6*, M1–M2); *p* < 0.01 (NS1: *Sox2* and *p75NTR*, M1–M2. NS2: *Pax6*, M2–M3; and *p75NTR*, M1–M2); *p* < 0.05 (NS1: *nestin*, M2–M3; *Pax6*, M1–M2 and M2–M3; *p75NTR*, M2–M3. NS2: *Sox2*, M1–M2; and *p75NTR*, M2–M3).

**Figure 4 biomedicines-13-01560-f004:**
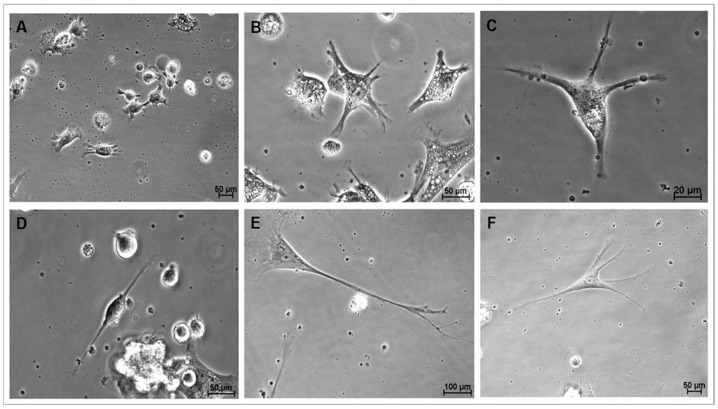
Representative photomicrographs of cells during directed differentiation. (**A**): Cells from group M1 on day 2 of culture showing filopodia and neurites (phase contrast, 10×/22× magnification); (**B**): Cells from group M1 on day 3 of culture displaying abundant cytoplasmic and membrane projections, and numerous cytoplasmic lipid droplets (phase contrast, 40×/0.55 magnification) (**C**): Cells from group M1 after 7 days in culture showing four thin projections of cytoplasm and elongating membrane (phase contrast 40×/0.55 magnification) (**D**): Cells from group M2 on day 7 of culture with neuroblast-like morphology (phase contrast, 40×/0.55 magnification) (**E**): Cells from group M1 on day 21 of culture, resembling a mature neuron, with visible soma, nucleus, dendritic arborization, and a fully elongated axon with terminal branches at its end, reminiscent of the synaptic terminals of fully differentiated neurons (phase contrast, 20×/0.40 magnification) (**F**): differentiated cells from group M1 on day 23 of culture with five thin cytoplasm and membrane projections (phase contrast 20×/0.40 magnification).

**Figure 5 biomedicines-13-01560-f005:**
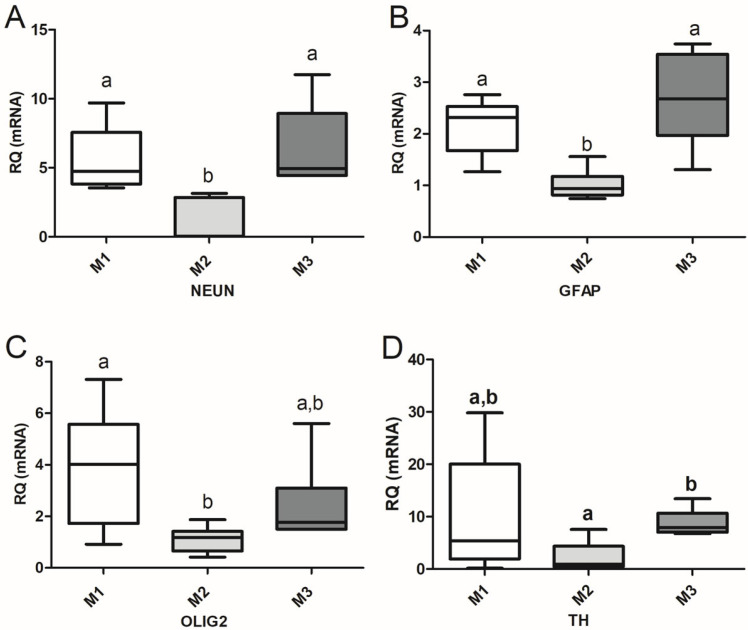
Graphical representation of the relative quantification (RQ) of genes characteristic of neural differentiation, *NeuN* (**A**; mature neuron); *GFAP* (**B**; astrocyte); *Olig2* (**C**; oligodendrocyte); and *TH* (**D**; dopaminergic precursors and neurons), during the directed differentiation culture of M1, M2 and M3 cells. The transcription of each gene in cells from the M2 group was used as the reference for RQ. Different letters indicate significant differences: *p* < 0.001 (*NeuN*; M1–M2, M2–M3; *GFAP*; M2–M3); *p* < 0.01 (*GFAP*, M1–M2; *Olig2*, M1–M2); *p* < 0.05 (*TH*; M1–M2).

**Figure 6 biomedicines-13-01560-f006:**
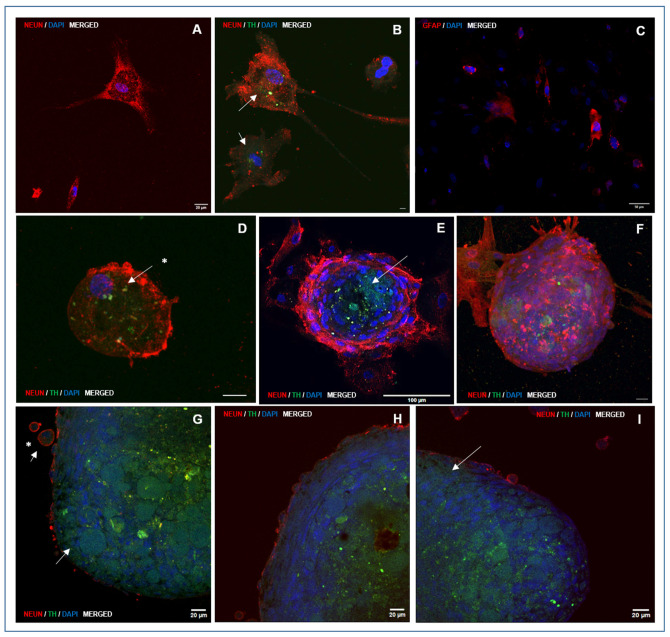
Representative photomicrographs of double immunolocalization of markers NeuN and TH, specifically expressed by differentiated neurons and dopaminergic neural precursors and neurons, respectively, at the end of culture for directed differentiation and GFAP. Panel 1: Immunolocalization of NeuN (red staining) in neurons differentiated on growth surface (**A**,**B**), in cells of NS-outer sheet (**E**–**I**), and in the cell membrane of neuron precursor cells arising from NS (**D**,**G**–**I**). GFAP was localized in sparsely distributed cells on the growth surface (**C**; red staining). TH was specifically immunolocalized in some morphologically differentiated neurons (**B**; white arrows), in core cells placed in NS generated during directed differentiation (**E**; white arrow), in cells from the outer sheet of medium and large NS (**G**–**I**; white arrows), and in neural precursor cells originating from NS during directed differentiation (**D**,**G**; white arrow and asterisks). Nuclei were counterstained with DAPI. (**A**,**B**,**D**,**G**–**I**: HC, PL APO CS2, 63×/1.40 magnification, oil); (**C**,**E**,**F**: HC, PL APO C52, 20×/0.75 magnification, dry). Images show combined immunolocalization of markers NeuN/TH/DAPI and GFAP/DAPI. The corresponding individual immunolocalization images are provided in Appendix A.

**Figure 7 biomedicines-13-01560-f007:**
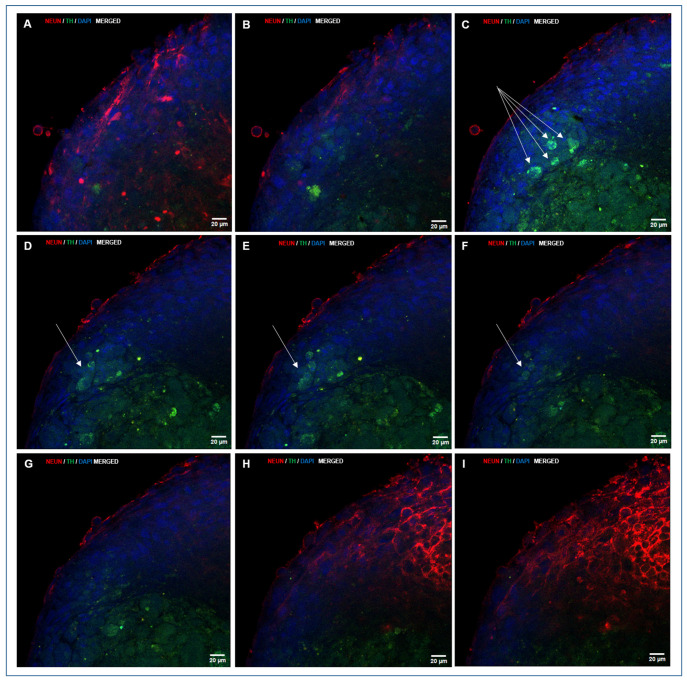
A sequence of confocal laser microscopy planes of an NS generated during directed differentiation, showing double immunolocalization of NeuN and TH. When visualizing images that start on the growth surface from a deep plane of the NS (**A**), NeuN staining (red) is still evident. Progression towards outer planes (**B**,**C**) shows that TH staining (green) is exhibited in NS core (**B**–**F**), and TH positive cells start to appear among cells from the outer layer (**C**, see four cells pointed by arrows), and are placed in different planes, as shown when progressing towards the mid-plane of the NS (**D**–**F**, see three more cells, pointed by an arrow), to finish in the outer sheet cover of cells that show mainly NeuN staining (red). Nuclei were counterstained with DAPI. (**A**–**I**, HC, PL APO CS2, 63×/1.40 magnification, oil).

**Figure 8 biomedicines-13-01560-f008:**
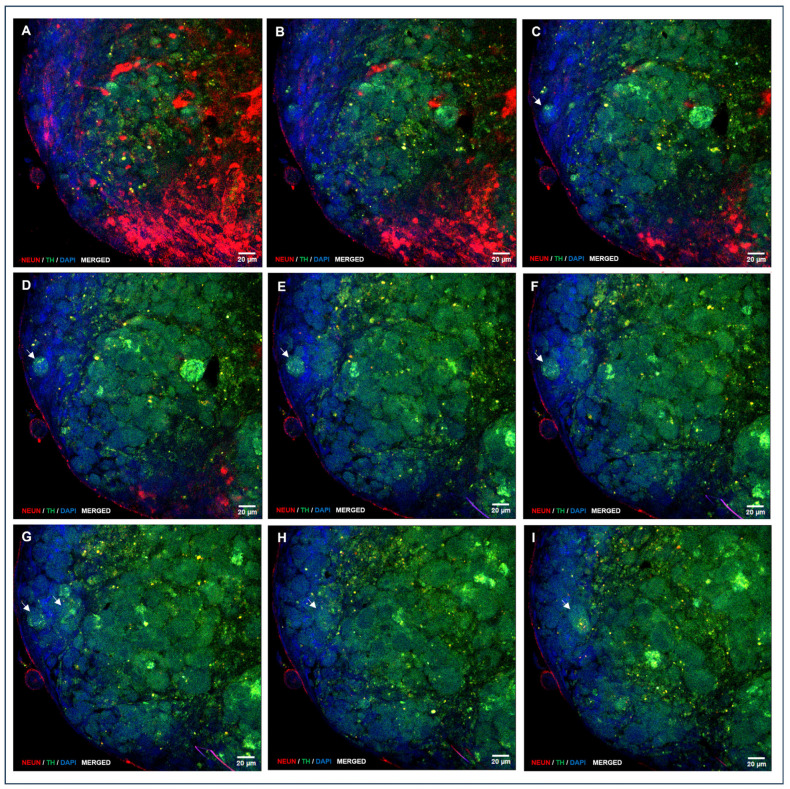
A sequence of confocal laser microscopy planes of an NS generated during directed differentiation, showing double immunolocalization of NeuN and TH. Imaging starts on the outer planes and progresses towards the inner planes. Cells from the outer sheet exhibit NeuN immunolocalization (**A**–**I**; red staining), and TH is mainly expressed in core cells that frequently come into the NS outer sheet (**C**–**I**; green staining see arrows). Nuclei were counterstained with DAPI. (**A**–**I**, HC, PL APO CS2, 63×/1.40 magnification, oil).

**Figure 9 biomedicines-13-01560-f009:**
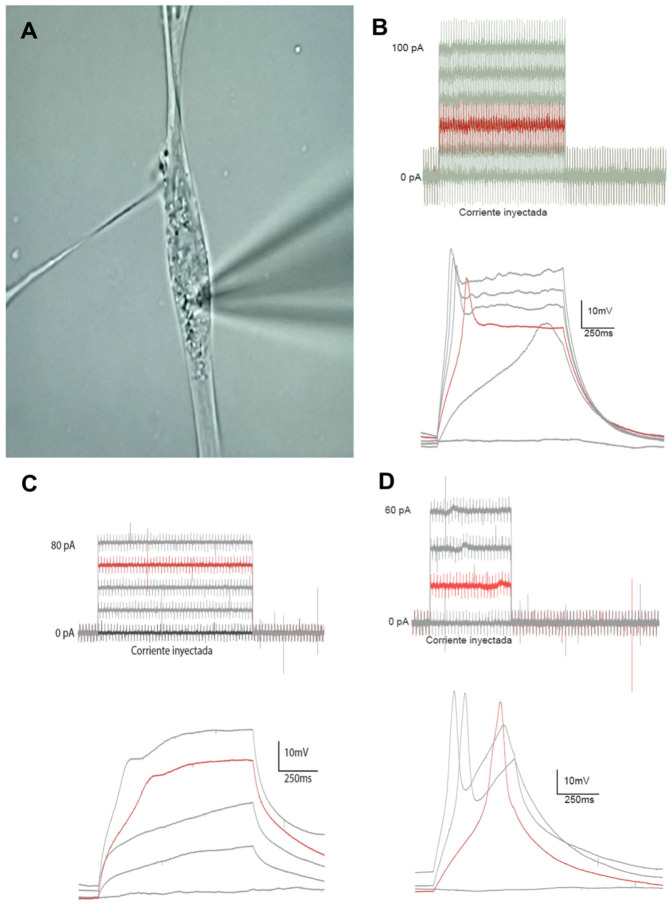
Recording electrical activity in neural cells differentiated from sheep OCC-NS, exposed to increasing 5 mA stimuli and characterized through patch-clamp analysis. (**A**) Photomicrograph showing the pipette placement on the membrane of one of the cells analyzed during the electrophysiological characterization. (**B**–**D**) Representative recordings of different injected currents and cellular responses recorded by patch-clamp analysis. Black traces represent the injected current threshold, while red traces correspond to the evoked action potential: plateau potential (**B**), passive electrical activity (**C**), and complete action potential (**D**).

**Table 1 biomedicines-13-01560-t001:** Primers used for quantitative Reverse Transcription Polymerase Chain Reaction of transcripts used to characterize neurospheres derived from sheep (*Ovis aries*) ovarian cortical cells.

Transcript	Accession Number	Primers
*Nestin*	XM_004002626.1	F CTCCAGAACTACTAAAGCCTACAG R CCAGCGACTCTTGACTTTCC
*Sox2*	X96997.1	F GGAGGACAGCAAGAAACAG R GCGTGAGTGTAGATGGGA
*Pax6*	XM_004016373.1	F AACATCCTTTACCCAAGAGCA R TTTCTCGGGCAAACACATCTG
*p75NTR*	XM_004013355.1	F CTCATCCCTGTCTATTGCTCCA R TTCCACCTCTTGAAGGCGA
*GFAP*	XM_004012992.1	F GAAGCAGATGAAGCCACCC R CTCAATCTTCCTCTCCAGATCCA
*Olig2*	XM_015091722.1	F GGCTTCAAGTCATCCTCGTC R GCTCCGTCATCTGTTTCTTGTC
*NeuN*	XM_042256257.1	F: GGCAAATGTTCGGGGTCAAC R: CATCCTGATACACGACCGCT
*TH*	XM_027959751.2	F: GTCTCTGAGCTGGACGCC R: GTGGGCTTCGAATGTCTCAAA

## Data Availability

The data presented in the current study are available on request from the corresponding author.

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
