# Peer review of "Proneurogenic Actions of FSH During Directed Differentiation of Neural Stem and Progenitor Cells from Ovarian Cortical Cells Towards the Dopaminergic Pathway"

_biomedicines, 2025, doi:10.3390/biomedicines13071560_

Round 1
Reviewer 1 Report
Comments and Suggestions for Authors
The research article entitled "Proneurogenic Actions of FSH during Directed Differentiation
of Neural Stem and Progenitor Cells from Ovarian Cortical
Cells towards the Dopaminergic Pathway" covered the generation of the dopaminergic neuron using specially designed differentiation medium. The present work incorporates scientific soundness along with its importance to the audience of the journal. There are some minor queries that need to be addressed before publication-
- The acronym TH should be explained in the Abstract before use.
- Why are the background in each figure 4(A-F) kept changing?
- Are the neurons shown in figures 4D and E are same?
- Did the author track the maturation of the group of neurons in Figure 4?
- Figure 6, panels 1 and 2 can not explain the part of the neuron scanned in these figures. Further, the author should keep scanning the same area while scanning for different immunolocalization.
Author Response
ANSWERS TO REVIEWER 1 (MS10.3390)
Reviewer 1: Comments and Suggestions for Authors
The research article entitled "Proneurogenic Actions of FSH during Directed Differentiation of Neural Stem and Progenitor Cells from Ovarian Cortical
Cells towards the Dopaminergic Pathway" covered the generation of the dopaminergic neuron using specially designed differentiation medium. The present work incorporates scientific soundness along with its importance to the audience of the journal. There are some minor queries that need to be addressed before publication-
1.- The acronym TH should be explained in the Abstract before use. Answer to referee: Thank you very much for this comment. TH acronym has been defined in the abstract of the revised manuscript as Tyrosine hydroxylase (line 44).
2.- Why are the background in each figure 4(A-F) kept changing? Answer to referee: Each microphotograph has an individual need of light and contrast, when being taken under the microscope, since most of our specimens are tridimensional structures combined with cells growing on a flat surface. In addition, we have phase contrast and plan achromat objectives. Nis elements Nikon Software allows uniformity of background which we had mislead. Now we submit these figures with the original grey backgound as Figure 1, and Figure 4, revised tiff files. Thank you very much for this comment.
3.- Are the neurons shown in figures 4D and E are same? Answer to referee: No, they aren´t. In fact, if referee looks at figure legend, 4D (between lines 442 and 443 of the revised manuscript) microphotograph points to a neuroblast-like cell found in M2 group (40X magnification, phase contrast), whereas 4E microphotograph shows a neuron-like differentiated cell from M1 group (20X magnification, phase contrast) with morphologically relevant features so as dendritic arborisation, and fully elongated axon, among others. Please, indicate us if this explanation solves doubts about this figure. Thanks to referee.
4.-Did the author track the maturation of the group of neurons in Figure 4? Answer to referee: In this particular research we did not track maturation of neurons since we are planning to conduct video time lapse for this purpose in future experiments, which is a much more reliable procedure to follow up neuron differentiation without interfering with cells growth in culture. We take microphotographs to our cells every 48 hours and cells are out the incubator for a very short time to avoid cell damage. For this reason, it is not technically possible to monitor single cell morphological differentiation with methods described in the current manuscript. We thank referee for this comment because we also see the need for monitoring this process in vitro, by video time lapse.
5.-Figure 6, panels 1 and 2 cannot explain the part of the neuron scanned in these figures. Further, the author should keep scanning the same area while scanning for different immunolocalization. Answer: We agree with referee, that we should have presented NeuN, TH, GFAP, DAPI staining by separate in different images, and then a merged one. To condense representative images and reduce space, we have only presented the merged images, with colour indications (red, green, blue) in the figure legend. But we should have indicated “merged” on each one or have presented by separate the individual immunostaining images and the merged one. We have prepared new figures to submit as supplementary material (Figure 1-Supplementary Material; Figure 2-Supplementary Material; Figure 3 Supplementary Material) where immunolocalization of each individual marker is presented after scanning the cells and neurospheres contained in pannel 1 of figure 6, and those from other neurospheres analysed in these experiments. Please, we have avoided to prepare panels with individual immunolocalization of neurospheres appearing in panels 2 and 3, because it represents an excess of images, and we can provide them under request. If referee thinks that this must be prepared unless as supplementary material, we will do it.
Accordingly with these changes, we have corrected figure legend 6 text (lines 503-517), and we have indicated that individual immunolocalization of each marker is provided in supplementary materials, Figure 1. On the other hand, neurons appear only in figure 6, panel 1 A, B. Astrocytes are shown in figure 6, panel 1 C. A dopaminergic neuron precursor is shown in figure 6 panel 1 F. The remaining images correspond to neurospheres. Should we apply these changes to all images, or should we include all individual immunolocalization images in supplementary material? Thanks to referee for this comment.
Submission Date
24 April 2025
Date of this review
14 May 2025 12:25:26
Reviewer 2 Report
Comments and Suggestions for Authors
This investigation has its novelty demonstrating the potential of ovarian cortical cell-derived neural stem/progenitor cells (OCC-NSPCs), enhanced by follicle-stimulating hormone (FSH), to generate functional dopaminergic neurons, representing a comprehensive and promising approach for autologous cell therapy in Parkinson’s disease. This study’s significance stems from the potential for autologous cell therapy for Parkinson’s disease, it provides new knowledge helping the understanding of the role of FSH in neurogenesis, and the demonstration of electrophysiological functionality. These findings contribute to the development of new therapeutic strategies also for other neurodegenerative diseases. I believe this work can be recommended. The following minor issues can be considered to improve discussion:
-
There are several specific safety concerns with pluripotent stem cell-derived therapies which include the risk of teratoma formation from residual undifferentiated cells, immune rejection requiring immunosuppression and its associated complications, and also the possibility of transplanted cells exhibiting unintended and potentially harmful effects within the brain. Adding discussion on this context in relation to target genes can be crucial. In particular:
-
The authors reported that cell proliferation and neurosphere development were larger in OCC-NSPCs from groups M1 (epidermal growth factor added) and M3 (FSH added) than in M2. This observation can be related to potential risk of tumorigenicity: increased cell proliferation, even during in vitro differentiation, raises the possibility that some cells may not differentiate properly and could retain their proliferative potential if transplanted. The greater proliferation in M1 and M3 compared to M2 is concerning.
-
In Discussion the authors wrote: “Moreover, the addition of FSH to the defined culture medium during the CEPs promotes neurogenic development, increasing the expression of transcripts characteristic of mature neural cells, including TH expression and its corresponding protein immunolocalization” lacks the quantitative precision necessary to convey the true impact of the finding. I recommend to enhance this statement by specifying the magnitude of that effect.
-
For clarity, please define the abbreviations EDH, FGF, and BDNF as you did for other abbreviations within the abstract.
Author Response
Comments and Suggestions for Authors
This investigation has its novelty demonstrating the potential of ovarian cortical cell-derived neural stem/progenitor cells (OCC-NSPCs), enhanced by follicle-stimulating hormone (FSH), to generate functional dopaminergic neurons, representing a comprehensive and promising approach for autologous cell therapy in Parkinson’s disease. This study’s significance stems from the potential for autologous cell therapy for Parkinson’s disease, it provides new knowledge helping the understanding of the role of FSH in neurogenesis, and the demonstration of electrophysiological functionality. These findings contribute to the development of new therapeutic strategies also for other neurodegenerative diseases. I believe this work can be recommended. The following minor issues can be considered to improve discussion:
- There are several specific safety concerns with pluripotent stem cell-derived therapies which include the risk of teratoma formation from residual undifferentiated cells, immune rejection requiring immunosuppression and its associated complications, and also the possibility of transplanted cells exhibiting unintended and potentially harmful effects within the brain. Adding discussion on this context in relation to target genes can be crucial. In particular:
ANSWER TO REFEREE: We agree with the referee that, despite the promising potential of pluripotent stem cells in regenerative medicine, it remains essential to develop patient-specific stem cells that do not entail ethical dilemmas, immunological incompatibilities, or risks of genetic instability. Such cells would provide a readily available and safer source for therapeutic applications. Within this context, adult stem cells derived from non-central nervous system tissues are gaining attention as a viable alternative. These adult stem cells offer several advantages over embryonic and foetal counterparts, particularly due to the absence of ethical and legal controversies. Moreover, they are widely distributed across almost all tissues in the human body and can be easily isolated, enabling autologous use and minimizing the risk of immune rejection [Bacakova et al., 2018].
In contrast to induced pluripotent stem cells (iPSC), adult stem cells typically do not present the tumorigenic potential associated with certain reprogramming strategies tested in animal models [Lee et al., 2013].
Nonetheless, as correctly emphasized by the referee, assessing tumorigenicity is a fundamental aspect of evaluating the safety of stem cell-based therapies. These evaluations are currently part of basic research and must precede preclinical studies. Ideal assessments should be rapid, sensitive, cost-effective, and scalable. Currently available methods include both in vivo and in vitro approaches, such as animal experimentation, soft agar assays, PCR analysis, flow cytometry, and microfluidic systems (Sato et al., 2019; Wang, 2023).
Accordingly, the following paragraph has been added to the revised manuscript (lines 73-84): “Despite the promising potential of pluripotent stem cells in regenerative medicine, there is a need to generate patient-specific SCs without ethical, immunological, and genetic instability concerns, providing an easily available and renewable source of cells with the potential to differentiate into fully functional DA neurons after transplantation [6]. In this context, adult SCs from non-central nervous system (CNS) tissues are emerging as a reliable alternative, offering advantages such as the absence of ethical or legal constraints, accessibility from body tissues, and the possibility of autologous use, thus preventing immune rejection [7,8]. Furthermore, these cells do not exhibit the tumorigenic risks associated with the use of induced pluripotent stem cell (iPSC) as observed in animal models [9]. Nevertheless, assessing tumorigenicity remains a critical step in ensuring the safety of stem cell-based therapies and is currently the focus of ongoing basic research preceding preclinical trials [9.10]”.
We have included the new references accordingly I the corresponding section:
- Bacakova, L.; Zarubova, J.; Travnickova, M.; Musilkova, J.; Pajorova, J.; Slepicka, P.; Kasalkova, N.S.; Svorcik, V.; Kolska, Z.; Motarjemi, H.; et al. Stem cells: Their source, potency and use in regenerative therapies with focus on adipose-derived stem cells—A review. Adv. 2018, 36, 1111–1126. https://doi.org/10.1016/j.biotechadv.2018.03.011
- Lee, A.S.; Tang, C.; Rao, M.S.; Weissman, I.L.; Wu, J.C. Tumorigenicity as a clinical hurdle for pluripotent stem cell therapies. Med. 2013, 19, 998–1004. https://doi.org/10.1038/nm.3267
- Sato, Y.; Bando, H.; Di Piazza, M.; Gowing, G.; Herberts, C.; Jackman, S.; Leoni, G.; Libertini, S.; MacLachlan, T.; McBlane, J.W.; Pereira Mouriès, L.; Sharpe, M.; Shingleton, W.; Surmacz-Cordle, B.; Yamamoto, K.; van der Laan, J.W. Tumorigenicity assessment of cell therapy products: The need for global consensus and points to consider. Cytotherapy 2019, 21, 1095–1111. https://doi.org/10.1016/j.jcyt.2019.10.001
- Wang, Z. Assessing Tumorigenicity in Stem Cell-Derived Therapeutic Products: A Critical Step in Safeguarding Regenerative Medicine. Bioengineering 2023, 10, 857. https://doi.org/10.3390/bioengineering10070857
- The authors reported that cell proliferation and neurosphere development were larger in OCC-NSPCs from groups M1 (epidermal growth factor added) and M3 (FSH added) than in M2. This observation can be related to potential risk of tumorigenicity: increased cell proliferation, even during in vitro differentiation, raises the possibility that some cells may not differentiate properly and could retain their proliferative potential if transplanted. The greater proliferation in M1 and M3 compared to M2 is concerning.
ANSWER TO REFEREE: Although cell proliferation may be one of the characteristics of tumorigenicity, it does not necessarily imply that the cells are tumorous. In fact, high proliferation and self-renewal activity, along with the capacity for differentiation, are properties that make these cells optimal candidates for use in regenerative medicine and cell therapy [Zakrzewski et al., 2019].
EGF and FGF2 are two neural-inducing factors secreted by cells located in the neurogenic niches of the brain. Their role is to reactivate quiescent neural stem cells and stimulate them to proliferate, generating a rich pool of cells capable to differentiating into neurons and glia to replace damaged brain cells. This process is characteristic of neurogenesis. As an essential biological process, it is not inherently associated with tumour induction. Nevertheless, the possibility exists, and we share the referee´s concern. This is a critical issue that will be addressed in future experiments, as previously mentioned.
We have included an observation in the discussion (lines 612 to 622), as follows: “EGF and FGF2, are neural inducing factors secreted in brain neurogenic niches that reactivate quiescent neural stem cells, promoting their proliferation and differentiation into neurons and glial cells to replace damaged cells. EGF and FGF2 exert these same actions as part of normal neurogenesis [48,49]. These effects are clearly reproduced in the experiments conducted in the current research, which is an indication that OCC-NSC should be further explored for their potential utility in autologous cell therapy. High proliferation, self-renewal, and differentiation abilities make SCs suitable for regenerative medicine and cell therapy [50]. However, as a possible side-effect, the tumour formation potential of these cells must be thoroughly investigated as part of the safety analyses, provided that EGF and FGF2 are also involved in tumour progression due to their proliferative actions on cancer cells [51]”.
The standardized defined media used for neuronal induction of human pluripotent stem cells and embryonic stem cells during the neurosphere assay typically contains both EGF and FGF2 (Bazán et al., 2004; Azari et al., 2010; Pan et al., 2016). Therefore, in the current experiment, both factors were included in the group used as a positive control (M1). This provides a reference for comparing the effects observed when the defined medium was supplemented with FSH (50 ng/ml).
We would like to address two points here: First, a pilot experiment with unpublished results has shown that FSH increases the secretion of FGF2 in these cultures. Together with our current results, this supports the hypothesis that FSH might upregulate the expression of EGF and FGF2 in neural stem cells in vitro, which could explain the similar responses observed in these two groups.
Second, the concentrations of FSH added to the culture medium (50 ng/ml) are comparable to physiological plasma concentrations and therefore reflect the usual concentrations to which NSC/NPC are exposed in the brain. In previous studies (Sánchez Maldonado et al., 2018) focused on general non-directed differentiation, cell proliferation during the differentiation stage was blocked by supplementing medium with antagonists of EGF and FGF2 receptors. Unfortunately, we could not use these antagonists in the current study, as the directed differentiation medium contains FGF8, which is essential for differentiation along the dopaminergic pathway.
We appreciate the referee´s valuable observation that we have included in the discussion section. We thank the referee for this important input. We have to incorporate these new references in the corresponding section.
- Zakrzewski, W.; Dobrzyński, M.; Szymonowicz, M.; Rybak, Z. Stem cells: past, present, and future. Stem Cell Res. Ther. 2019, 10, 68. https://doi.org/10.1186/s13287-019-1165-5
- Bazán, E.; Alonso, F.J.M.; Redondo, C.; López-Toledano, M.A.; Alfaro, J.M.; Reimers, D.; et al. In vitro and in vivo characterization of neural stem cells. Histopathol. 2004, 19, 1261–1275.
- Azari, H.; Rahman, M.; Sharififar, S.; Reynolds, B.A. Isolation and expansion of the adult mouse neural stem cells using the neurosphere assay. Vis. Exp. 2010, 45, e2393. https://doi.org/10.3791/2393
- Pan, W.K.; Yu, H.; Wu, A.L.; Gao, Y.; Zheng, B.J.; Li, P.; Yang, W.L.; Huang, Q.; Wang, H.J.; Ge, X. Combination of basic fibroblast growth factor and epidermal growth factor enhances proliferation and neuronal/glial differentiation of postnatal human enteric neurosphere cells in vitro. Neuroreport 2016, 27, 858–863. https://doi.org/10.1097/WNR.0000000000000626
- Sánchez-Maldonado, B.; Galicia, M.L.; Rojo, C.; González-Gil, A.; Flor-García, M.; Picazo, R.A. Spheroids spontaneously generated in vitro from sheep ovarian cortical cells contain integrating cells that exhibit hallmarks of neural stem/progenitor cells. Stem Cells Dev. 2018, 27, 1557–1576. https://doi.org/10.1089/scd.2018.0053
- Christie KJ, Turnley AM. Regulation of endogenous neural stem/progenitor cells for neural repair—factors that promote neurogenesis and gliogenesis in the normal and damaged brain. Front Cell Neurosci. 2013; 6:70. doi:10.3389/fncel.2012.00070. PMID: 23346046; PMCID: PMC3548228.
- Woodbury ME, Ikezu T. Fibroblast growth factor-2 signaling in neurogenesis and neurodegeneration. J Neuroimmune Pharmacol. 2014;9(2):92-101. doi:10.1007/s11481-013-9501-5. PMID: 24057103; PMCID: PMC4109802.
- Soeda A, Inagaki A, Oka N, Ikegame Y, Aoki H, Yoshimura S, Nakashima S, Kunisada T, Iwama T. Epidermal growth factor plays a crucial role in mitogenic regulation of human brain tumor stem cells. J Biol Chem. 2008;283(16):10958-10966.
- In Discussion the authors wrote: “Moreover, the addition of FSH to the defined culture medium during the CEPs promotes neurogenic development, increasing the expression of transcripts characteristic of mature neural cells, including TH expression and its corresponding protein immunolocalization” lacks the quantitative precision necessary to convey the true impact of the finding. I recommend to enhance this statement by specifying the magnitude of that effect.
ANSWER TO REFEREE: We appreciate the referee´s insightful comment and fully agree that the original statement lacked the quantitative precision necessary to convey the actual magnitude of the observed effect. In response, we have revised this paragraph to include specific quantitative data supporting our conclusions. The original paragraph has been replaced with this one (lines 584-599): “The results of this study, confirm previous contributions regarding the proneurogenic actions of FSH on OCC-NSPCs (16), by demonstrating again that this hormone increases NSPCs proliferation and self-renewal, and provide further progress on defining these actions by showing that priming NSPCs with FSH during CEPs, up-regulates the expression of transcripts specifically associated with terminal neuronal differentiation so as NeuN and, remarkably, TH (transcript and protein) after directed differentiation of these cells towards the DA pathway. Our results show that FSH (M3) up-regulates the mean NeuN expression be-tween two to three times with respect with control (M2), similarly than in positive control group (M1). Mean TH transcription of differentiated cells primed with FSH during CEPs (M3), increases up to three to four times the expression quantified in control (M2), with values slightly larger but non statistically significant with respect to positive control group (M2). Outstandingly, the actions of FSH on TH transcription during differentiation are neatly reflected in the percentages of NS cells exhibiting TH immunolocalization, with the largest percentages quantified after DA differentiation of NSPCs primed with FSH in CEPs (33.5%) over values of control (23.74%; M2), and not significantly larger than positive control group (M1; 29.35%; M1). Several other aspects of these results will be further discussed at the end of this section.
- For clarity, please define the abbreviations EDH, FGF, and BDNF as you did for other abbreviations within the abstract.
ANSWER TO REFEREE: We believe the reviewer is referring to EGF, FGF, and BDNF. We agree, and we have now defined these abbreviations in the abstract section as follows: EGF (epidermal growth factor), FGF (fibroblast growth factor), and BDNF (brain-derived neurotrophic factor).
Reviewer 3 Report
Comments and Suggestions for Authors
The article presents results of ovarian cells SC transforming to neuron cells producing dopamine.
The interesting data should be highlighted and clearly discussed in comparison with previous works as that in reference 12.
Introduction should include more data regarding the interest for studying the electrophysiological profile of studied cells. Also, data regarding survival of cells among neurons is desirable.
Discussion should be enriched, regarding particular effects of FSH, differences with previous works and the potential of studies in the membrane channels for neurodegenerative treatment. The role of additional enzymes to TH could be added as interesting to be applied in parkinson disease or other neurodegenerative diseases.
Conclusions should be clearly supported and different of some other reported findings for your group.
If well there are no plagiarism, the edition to improve presentation avoid the perception of very similar work carried out in previous works.
Author Response
Comments and Suggestions for Authors
The article presents results of ovarian cells SC transforming to neuron cells producing dopamine.
- The interesting data should be highlighted and clearly discussed in comparison with previous works as that in reference 12.
ANSWER TO REFEREE: We thank the reviewer for this valuable comment. In response, we have revised the Discussion section to highlight the novel aspects of the current study compared to our previous experiment (González-Gil et al., 2024). While our previous study demonstrated that FSH promotes self-renewal and upregulates the expression of early neuronal markers such as Nestin and Pax6 and the neuron precursor transcript DCX in OCC-derived neurospheres, the current study significantly expands these findings by demonstrating, for the first time, that FSH priming enhances the directed differentiation of these cells toward the DA lineage. Importantly, we report increased TH expression at both gene expression and immunolocalization, as well as functional neuronal activity assessed by patch-clamp recordings. Moreover, the combination of FSH with a defined differentiation medium, including SHH, FGF8, and BDNF provides a unique approach to obtain functionally mature DA neurons, highlighting the potential relevance of this protocol for neurodegenerative diseases such as PD.
According to referee, we have highlighted and discussed the interesting data in the current study comparing with previous experiments. We have added the following paragraph in the Discussion section (lines 584-599): “This is the first research in which FSH priming during CEPs of the NS assay is combined with directed differentiation of NSPCs towards DA pathway by using a defined me-dium containing SHH, FGF8, and BDNF, a procedure with possible eventual clinical implications for Parkinson's disease therapy.
The results of this study, confirm previous contributions regarding the proneurogenic actions of FSH on OCC-NSPCs (16), by demonstrating again that this hormone increases NSPCs proliferation and self-renewal, and provide further progress on defining these actions by showing that priming NSPCs with FSH during CEPs, up-regulates the expression of transcripts specifically associated with terminal neuronal differentiation so as NeuN and, remarkably, TH (transcript and protein) after directed differentiation of these cells towards the DA pathway. Our results show that FSH (M3) up-regulates the mean NeuN expression be-tween two to three times with respect with control (M2), similarly than in positive control group (M1). Mean TH transcription of differentiated cells primed with FSH during CEPs (M3), increases up to three to four times the expression quantified in control (M2), with values slightly larger but non statistically significant with respect to positive control group (M2). Outstandingly, the actions of FSH on TH transcription during differentiation are neatly reflected in the percentages of NS cells exhibiting TH immunolocalization, with the largest percentages quantified after DA differentiation of NSPCs primed with FSH in CEPs (33.5%) over values of control (23.74%; M2), and not significantly larger than positive con-trol group (M1; 29.35%; M1). Several other aspects of these results will be further discussed at the end of this section.”
- Introduction should include more data regarding the interest for studying the electrophysiological profile of studied cells. Also, data regarding survival of cells among neurons is desirable.
ANSWER TO REFEREE: We once again agree with the referee's comment. In response, we have added the following paragraph to the Introduction section to further emphasize the importance of studying the electrophysiological profile of the studied cells (lines 128-136): “Patch-clamp recording provides critical insight into the functional maturation and potential disease-related abnormalities of neurons. This approach is essential to ensure the functional identity and maturity of differentiated neurons, for their potential use in cell replacement therapies and drug discovery platforms. The patch-clamp technique provides high-resolution data of ion channel activity, action potential dynamics, synaptic currents, and membrane properties, allowing us to assess the functional consequences of PD-related changes at the cellular level. This provides insights for under-standing the complex pathophysiology of the disease and supports the development of effective therapeutic strategies and biomarkers [36].
We appreciate the reviewer’s suggestion that data regarding the survival of cells among neurons is desirable. In the new version of the manuscript, we have included a brief discussion on the relevance of assessing cell survival among differentiated neurons, highlighting the variability in survival rates and the need to optimize differentiation protocols to enhance therapeutic potential (lines137-140): “On the other hand, assessing the survival of differentiated DA neurons is crucial for evaluating the efficacy of SC-based therapies. A recent systematic review reported that graft survival rates in preclinical PD models vary widely, with a mean survival of 51% of transplanted cells [37]”.
And we have incorporated these references accordingly in the corresponding section.
- Grasselli, S.; Andolfi, A.; Di Lisa, D.; Pastorino, L. In vitro electrophysiological characterization of Parkinson's disease: challenges, advances, and future directions. Front. Neurosci. 2025, 19, 1584555. https://doi.org/10.3389/fnins.2025.1584555
- Comini, G.; Dowd, E. A systematic review of progenitor survival and maturation in Parkinsonian models. Neural Regen. Res. 2025, 20, 3172–3178.
- Discussion should be enriched, regarding particular effects of FSH, differences with previous works and the potential of studies in the membrane channels for neurodegenerative treatment.
ANSWER TO REFEREE: As mentioned in our first answer, we have highlighted and discussed the key findings of the current study by comparing them with previous experiments. Additionally, we appreciate and share the reviewer’s suggestion to further elaborate on the relevance of studying ion channels in the context of neurodegenerative disease. In response, we have added the following paragraph to the Discussion section (lines 753-762): This contribution offers the possibility to set up more complex studies related to altered ion channels function as underlying or associated pathophysiological mechanisms in neurodegenerative diseases, using this model. Studies on membrane ion channels provide promising therapeutic insights for neurodegenerative diseases. Ion channels play a central role in regulating neuronal excitability, synaptic activity, and intracellular calcium signalling, processes that are often dysregulated in diseases such as Parkinson’s [79,90]. Therefore, evaluating the electrophysiological properties of differentiated neurons, for instance through patch-clamp recordings, provides essential information on their functional identity and therapeutic potential [81,82].
We have included the new references in the corresponding section.
- Wang Z, Lin Y, Liu W, Kuang P, Lao W, Ji Y, Zhu H. Voltage-gated sodium channels are involved in cognitive impairments in Parkinson's disease-like rats. Neuroscience. 2019;418:231-243. doi:10.1016/j.neuroscience.2019.08.024. PMID: 31473280.
- Chen X, Feng Y, Quinn RJ, Pountney DL, Richardson DR, Mellick GD, Ma L. Potassium channels in Parkinson's disease: potential roles in its pathogenesis and innovative molecular targets for treatment. Pharmacol Rev. 2023;75(4):758-788. doi:10.1124/pharmrev.122.000743. PMID: 36918260.
- Li, Y.; Fu, J.; Wang, H. Advancements in Targeting Ion Channels for the Treatment of Neurodegenerative Diseases. Pharmaceuticals 2024, 17, 1462. https://doi.org/10.3390/ph17111462
- Orfali, R.; Alwatban, A.Z.; Orfali, R.S.; Lau, L.; Chea, N.; Alotaibi, A.M.; Nam, Y.W.; Zhang, M. Oxidative stress and ion channels in neurodegenerative diseases. Front. Physiol. 2024, 15, 1320086.
- The role of additional enzymes to TH could be added as interesting to be applied in Parkinson disease or other neurodegenerative diseases.
ANSWER TO REFEREE: We appreciate the referee's suggestion regarding the potential role of additional enzymes beyond tyrosine hydroxylase (TH) in the context of Parkinson's disease and other neurodegenerative disorders. We agree that considering complementary enzymes could enhance therapeutic approaches and provide a more comprehensive understanding of dopamine biosynthesis regulation under neurodegenerative conditions.
In the revised version of the manuscript, we have added a brief paragraph in the Discussion section highlighting how other key enzymes in the dopamine biosynthetic pathway, such as DOPA decarboxylase (DDC) and dopamine β-hydroxylase (DBH), may serve as complementary or synergistic therapeutic targets. The added paragraph (lines 688-695) reads: “It is of interest to explore in this culture system, the reliability of analyzing the expression and localization of other enzymes involved in catecholamine biosynthesis, such as aromatic L-amino acid decarboxylase and DA β-hydroxylase. Recent studies suggest that dysregulation of these enzymes contributes to the selective vulnerability of DA neurons, whereby it, may serve as complementary therapeutic targets in PD. This would also aid to generate strategies that modulate multiple enzymes in the DA pathway contributing to a greater understanding of the mechanisms involved PD pathogenesis [65].”.
Once again, we thank you for your valuable comment, which has helped strengthen the manuscript.
We incorporate the new references in the corresponding section:
- Wimalasena, K.; Adetuyi, O.; Eldani, M. Metabolic energy decline coupled dysregulation of catecholamine metabolism in physiologically highly active neurons: implications for selective neuronal death in Parkinson's disease. Front. Aging Neurosci. 2024, 16, 1339295. https://doi.org/10.3389/fnagi.2024.1339295
- Conclusions should be clearly supported and different of some other reported findings for your group.
ANSWER TO REFEREE: We appreciate the referee’s comment and agree that it is essential to clearly distinguish the conclusions of this study from our previous work. In response, we have revised the Conclusion section to explicitly highlight the novel aspects of this study as follow (lines 811-825): “This study establishes the experimental basis for the generation of DA neuron precursors and mature neurons from OCCs-NSPCs. It demonstrates that priming these cells with FSH during CEPs of the NS assay increases the expression of neural differentiation transcripts, similarly to the neural induction factors EGF and FGF2, and particularly TH expression and its corresponding protein, the enzyme responsible for converting tyrosine into L-DOPA, a key step in dopamine biosynthesis. A significant proportion of the differentiated cells at the end of the culture period exhibited functional electrophysiological responses, including plateau and action potentials, confirming their neuronal identity. Importantly, this study advances previous findings by reporting for the first time the combined use of FSH during NSPC expansion, with a previously established directed differentiation procedure to drive these cells towards the DA pathway. Future research will aim to refine the NS assay to generate a more homogeneous and abundant population of DA neurons ensuring the selection of functionally relevant neuron phenotypes and to perform comprehensive analyses that will help to elucidate whether NSPCs can be safely used in eventual autologous cell therapy. The potential application of these cells in neurodegenerative disease modelling and autologous cell therapy trials is currently under consideration”.
- If well there are no plagiarism, the edition to improve presentation avoid the perception of very similar work carried out in previous works.
ANSWER TO REFEREE: We thank the reviewer for this comment. We agree that clear and careful presentation is essential to distinguish this study from our previous publications. We have therefore revised the manuscript to improve its clarity and emphasize novel aspects of the present study, particularly in the Introduction, Discussion, and Conclusion sections.
These revisions highlight the functional validation of the identity of dopaminergic neurons and the novel combination of FSH with defined differentiation factors, aspects not addressed in our previous study.
.
Round 2
Reviewer 3 Report
Comments and Suggestions for Authors
The manuscript is improved, it is suitable to be published in this form. Thank you for considering all my comments.